# Safety and Survival Outcomes of Liver Resection following Triple Combination Conversion Therapy for Initially Unresectable Hepatocellular Carcinoma

**DOI:** 10.3390/cancers15245878

**Published:** 2023-12-17

**Authors:** Yin Long, Jue Huang, Jianguo Liao, Dongbo Zhang, Ziqi Huang, Xiaodong He, Lei Zhang

**Affiliations:** 1Department of Hepatobiliary Surgery, Sun Yat-sen Memorial Hospital, Sun Yat-sen University, Guangzhou 510120, China; longy63@mail2.sysu.edu.cn (Y.L.); huangj588@mail2.sysu.edu.cn (J.H.); liaojg5@mail2.sysu.edu.cn (J.L.); huangzq39@mail2.sysu.edu.cn (Z.H.); hohiot@mail2.sysu.edu.cn (X.H.); 2Department of Breast Surgery, Sun Yat-sen Memorial Hospital, Sun Yat-sen University, Guangzhou 510120, China; zhangdb5@mail.sysu.edu.cn

**Keywords:** hepatocellular carcinoma, conversion therapy, liver resection, unresectable

## Abstract

**Simple Summary:**

This study aimed to investigate the safety and long-term survival outcomes of salvage liver resection in patients with initially unresectable hepatocellular carcinoma (HCC) who had undergone triple combination conversion therapy (TACE/HAIC + TKIs + ICIs). While hepatectomy following conversion therapy exhibited elevated incidence rates of intra-abdominal bleeding, biliary leakage, post-hepatectomy liver failure, and Clavien–Dindo grade IIIa complications compared to pure hepatectomy, their safety profiles remained acceptable when appropriate medical interventions were administered. Patients with initially unresectable HCC, who successfully underwent conversion resection, demonstrated comparable overall survival (OS) and recurrence-free survival (RFS) to patients with initially resectable HCC.

**Abstract:**

Triple combination conversion therapy, involving transcatheter arterial chemoembolization (TACE) or hepatic arterial infusion chemotherapy (HAIC) combined with tyrosine kinase inhibitors (TKIs) and immune checkpoint inhibitors (ICIs), has shown an encouraging objective response rate (ORR) and successful conversion surgery rate in initially unresectable hepatocellular carcinoma (HCC). However, the safety and long-term survival outcomes of subsequent liver resection after successful conversion still remain to be validated. From February 2019 to February 2023, 726 patients were enrolled in this retrospective study (75 patients received hepatectomy after conversion therapy [CLR group], and 651 patients underwent pure hepatectomy [LR group]). Propensity score matching (PSM) was used to balance the preoperative baseline characteristics. After PSM, 68 patients in the CLR group and 124 patients in the LR group were analyzed, and all the matching variables were well-balanced. Compared with the LR group, the CLR group experienced longer Pringle maneuver time, longer operation time, and longer hospital stays. In addition, the CLR group had significantly higher incidence rates of intra-abdominal bleeding, biliary leakage, post-hepatectomy liver failure (PHLF), and Clavien–Dindo grade IIIa complications than the LR group. There were no significant statistical differences in overall survival (OS) (hazard ratio [HR] 0.724; 95% confidence interval [CI] 0.356–1.474; *p* = 0.374) and recurrence-free survival (RFS) (HR 1.249; 95% CI 0.807–1.934; *p* = 0.374) between the two groups. Liver resection following triple combination conversion therapy in initially unresectable HCC may achieve favorable survival outcomes with manageable safety profiles; presenting as a promising treatment option for initially unresectable HCC.

## 1. Introduction

Hepatocellular carcinoma (HCC), the most common type of primary liver cancer, ranks as the third leading cause of cancer-related deaths worldwide [1]. Liver resection provides the best chance for long-term survival in early-stage HCC patients (5-year survival rates of 60–80%) [2]. However, the majority of HCC patients are diagnosed at an intermediate or advanced stage, thus missing the opportunity for curative treatment to achieve long-term survival [3]. Previous studies reported that a portion of initially unresectable HCC could achieve tumor shrinkage and downstaging after conversion therapies such as transcatheter arterial chemoembolization (TACE), radiotherapy, and chemotherapy, subsequently regaining the opportunity for curative resection [4,5,6]. Nevertheless, the development of conversion therapy was limited by the low objective response rate (ORR) and low successful conversion surgery rate.

In recent years, the development of tyrosine kinase inhibitors (TKIs) and immune checkpoint inhibitors (ICIs), as well as the exploration of diverse combination treatment strategies, have brought about new breakthroughs in conversion therapy. The combination of TKIs and ICIs could achieve an ORR of approximately 30% in advanced HCC [7,8]. In addition, the triple combination of locoregional therapies such as TACE, or hepatic arterial infusion chemotherapy (HAIC), with TKIs and ICIs has shown potential in further improving the ORR (approximately 60%) and successful conversion surgery rate (20%~40%) in the treatment of initially unresectable HCC [9,10,11,12]. Consequently, an increasing number of initially unresectable HCC are now being offered the opportunity for curative surgical resection once again. However, the use of combination therapies may lead to increased severe adverse events (AEs), severe postoperative complications, and greater difficulty of the operation, which undoubtedly raises concerns among surgeons regarding the implementation of sequential liver resection [13].

Therefore, this study was conducted to investigate the safety and efficacy of surgical resection following triple combination conversion therapy with TACE/HAIC + TKIs + ICIs in patients with initially unresectable HCC.

## 2. Materials and Methods

### 2.1. Patient Selection

A total of 1239 patients with HCC who underwent pure liver resection or salvage liver resection after triple combination conversion therapy between February 2019 and February 2023 at Sun Yat-sen Memorial Hospital were screened for study eligibility. The exclusion criteria for this study were as follows: (1) patients aged < 18 years or aged > 75 years; (2) HCC patients with extrahepatic metastasis; (3) patients with recurrent HCC, or combined hepatocellular-cholangiocarcinoma confirmed by histopathological examination; (4) patients with HCC concurrent with other malignant tumors; (5) patients who underwent combined surgery on other vital organs during the same operation; (6) patients who received associating liver partition and portal vein ligation for staged hepatectomy (ALPPS) or portal vein embolization (PVE) for insufficient remnant liver volume; and (7) patients with insufficient or unavailable data. Finally, 726 patients (75 patients received hepatectomy after conversion therapy [CLR group], and 651 patients underwent pure hepatectomy [LR group]) were selected. The informed consent of patients was waived due to the retrospective study design. This study was approved by the institutional review board of Sun Yat-sen Memorial Hospital (SYSKY-2023-854-01).

### 2.2. Conversion Therapy

The patients with the following characteristics were considered initially unresectable and suitable for conversion therapy: (1) insufficient remnant liver volume; (2) tumor close to the crucial intrahepatic vessels leading to unfeasible R0 resection; (3) tumor numbers ≥ 4, or patients with macrovascular invasion; and (4) tumor deemed unresectable by two or more associate chief physicians or higher-ranked hepatobiliary surgeons.

For patients considered initially unresectable and suitable for conversion therapy, a triple combination therapy approach, which involved locoregional therapies (TACE or HAIC) plus TKIs and ICIs, could be used. It is important to note that the conversion therapy regimens were not fixed due to the varying circumstances of individual patients. TACE or HAIC procedures were performed by experienced interventional radiologists at Sun Yat-sen Memorial Hospital. TACE: With the assistance of a digital subtraction angiography system, a microcatheter was super-selected to each vascular branch of the tumor, and a chemotherapy solution, consisting of a mixture of epirubicin and iodized oil, was slowly injected until the tumor lesion was adequately saturated with the iodized oil. Subsequently, gelatin sponge particles were injected, as needed, until there was evident stasis of blood flow at the tip of the microcatheter. HAIC was performed on the FOLFOX regimen: a 3 h arterial infusion of 85 mg/m^2^ of oxaliplatin on day 1, 2 h arterial infusion of 400 mg/m^2^ of leucovorin on day 1, and a bolus injection of 400 mg/m^2^ of fluorouracil on day 1, followed by fluorouracil infusion of 2400 mg/m^2^ for 46 h every 3 weeks [14]. The drug dosages were subject to adjustment based on tumor condition, liver function grading, and chemotherapy tolerance. TKIs such as lenvatinib (8 mg/day), donafenib (200 mg, twice daily), and apatinib (250 mg/day), as well as ICIs such as pembrolizumab (200 mg, every 3 weeks), camrelizumab (200 mg, every 3 weeks), sintilimab (200 mg, every 3 weeks), and tislelizumab (200 mg, every 3 weeks), were used. To ensure the safety of the surgical procedure, it was common practice to discontinue TACE/HAIC approximately 4 weeks before surgery [15]. TKIs were typically stopped 1 to 2 weeks before surgery, while ICIs were usually discontinued around 2 to 4 weeks prior to surgery [15]. Patient assessments for resectability were conducted every 1–2 months.

Criteria for resectability after conversion therapy included (1) a Child–Pugh score ≤ 7 points, an indocyanine green retention rate at 15 min (ICG R15) ≤ 20%, and an Eastern Cooperative Oncology Group performance status score of 0–1; (2) patients without liver cirrhosis having a remnant liver volume ≥ 30% of standard liver volume, or patients with liver cirrhosis having a remnant liver volume ≥ 45% of standard liver volume; (3) an assessment of tumor lesions showing complete response (CR) or partial response (PR), or stable disease (SD) persisting for 3–4 months based on the modified Response Evaluation Criteria in Solid Tumors (mRECIST) criteria; (4) computed tomography (CT) or magnetic resonance imaging (MRI) indicating inactivation and regression of vascular tumor thrombi, surgical margin ≥ 1.0 cm, with an aim to achieve R0 resection; and (5) absence of other contraindications for surgery.

### 2.3. Data Collection and Follow-Up

Preoperative baseline data were collected, including gender, age, HBV virology status, HCV virology status, liver cirrhosis, Child–Pugh score, tumor location, number of tumors, maximum tumor diameter, surgical approach, platelet (PLT) count, international normalized ratio (INR), prothrombin time (PT), alanine aminotransferase (ALT) level, aspartate transaminase (AST) level, albumin (ALB) level, total bilirubin (TBil) level, alpha-fetoprotein (AFP) level, and Barcelona Clinic Liver Cancer (BCLC) stage. Perioperative outcomes (blood loss, blood transfusion, hepatic inflow occlusion, operation time, Intensive Care Unit (ICU) stay, hospital death, and hospital stays) and postoperative complications (including intra-abdominal bleeding, biliary leakage, post-hepatectomy liver failure (PHLF), etc.) were collected to compare the surgical safety of the CLR group and the LR group. The therapeutic efficacy was compared by assessing the overall survival (OS) and recurrence-free survival (RFS) between the two groups. The incidence of postoperative biliary leakage and PHLF was evaluated according to the International Study Group for Liver Surgery criteria [16,17]. The severity of postoperative complications was graded based on the Clavien–Dindo classification system [18]. In order to obtain additional information, the postoperative pathological results of both groups were also compared.

After discharge, patients underwent CT scans or MRI every 3–6 months for follow-up, in addition to being monitored for tumor markers such as AFP and protein induced by vitamin K absence or antagonist-II (PIVKA-II). In case of suspicion of HCC recurrence based on tumor marker levels or imaging findings, treatment measures such as TACE, local ablation, liver resection, or liver transplantation could be considered depending on the specific circumstances. RFS refers to the time from the patient undergoing surgical treatment to the first tumor recurrence, death, or the last follow-up date. OS was defined as the time from the operation to death due to any cause, or the last follow-up date.

### 2.4. Statistical Analysis

Continuous variables were represented using the median (interquartile range, IQR). The Student’s *t*-test or Mann–Whitney U test was employed for comparing continuous variables. Categorical variables were represented as numbers (percentages). For comparing categorical variables, Pearson’s chi-square test, Fisher’s exact test, or Mann–Whitney U test was utilized. The Kaplan–Meier method was used to generate and compare survival curves. The hazard ratios (HRs) with 95% confidence intervals (CIs) for OS and RFS were estimated using the Cox proportional hazards model. Statistical significance was defined as a two-sided *p*-value < 0.05. All data analyses were conducted using R (version 4.2.3) and SPSS 26.0 software.

To reduce the influence of confounding factors and selection bias, propensity score matching (PSM) was employed to ensure the statistical comparability of baseline characteristics between the two groups. This study utilized multivariable logistic regression analysis to calculate propensity scores for each patient, taking into account factors such as gender, age, HBV virology status, HCV virology status, liver cirrhosis, Child–Pugh score, tumor location, number of tumors, maximum tumor diameter, surgical approach, AFP level, and BCLC stage. R software (version 4.2.3) was then utilized to perform the nearest neighbor matching without replacement at 1:2 ratio, with a caliper value set at 0.2. Subgroup analyses based on age, sex, liver cirrhosis, tumor number, tumor size, AFP, and BCLC stage were conducted to evaluate the treatment effects. Standardized mean difference (SMD) was used to evaluate the balance of variables between groups, with SMD values less than 0.1 indicating very small differences; 0.1–0.3 indicating small differences; and values higher than 0.3 indicating moderate-large differences.

## 3. Results

### 3.1. Baseline Characteristics and Conversion Therapy Characteristics

A total of 726 HCC patients who underwent liver resection were included in the analysis, with 75 patients in the CLR group and 651 patients in the LR group. The preoperative baseline characteristics of the two groups before and after PSM are illustrated in Table 1. Compared with the LR group, the CLR group was significantly younger (median [IQR], 53 [43–59] years vs. 55 [48–64] years; *p* = 0.015), and had more HBV infection rate (93.3% vs. 82.3%, *p* = 0.024), a higher occurrence of multiple tumors (54.7% vs. 23.3%; *p* < 0.001), bigger tumor size (median [IQR], 7.4 [4.3–9.7] cm vs. 5.0 [3.0–8.5] cm; *p* = 0.038), and more advanced BCLC stage (0/A/B/C, 0/28.0%/24.0%/48.0% vs. 9.1%/55.6%/16.7%/18.6%; *p* < 0.001). After PSM, 68 patients in the CLR group and 124 patients in the LR group were analyzed, and all the matching variables were well-balanced (Table 1, Appendix A). The conversion therapy regimen and details of the 68 patients is shown in Table 2. Of the patients, 16 (23.5%) received TACE (median [IQR], 2 [2,3,4]) and 52 (76.5%) received HAIC (median [IQR], 4 [3,4]). In addition, the most commonly used TKIs + ICIs regimen was lenvatinib + camrelizumab (28.0%), followed by donafenib + tislelizumab (17.6%), lenvatinib + pembrolizumab (16.2%), apatinib + camrelizumab (16.2%), lenvatinib + tislelizumab (13.2%), lenvatinib + sintilimab (4.4%), and donafenib + camrelizumab (4.4%). The median time of TKI treatment was 100 (IQR, 92–164) days, and the median ICI treatment cycle was 4 (IQR, 3–4). Furthermore, the median duration of conversion therapy was 126 (IQR, 92–164) days. Based on the mRECIST criteria, 18 (26.5%) patients achieved CR, 45 (66.2%) had PR, and 5 (7.3%) were considered to have SD after conversion therapy.

### 3.2. Intraoperative and Postoperative Outcomes

Table 3 shows the intraoperative and postoperative data of the two groups after PSM. Patients in the CLR group had significantly longer Pringle maneuver time (median [IQR], 45 [11–76] min vs. 30 [0–55] min; *p* = 0.013), longer operation time (median [IQR], 268 [215–349] min vs. 230 [180–300] min; *p* = 0.037), and longer hospital stays (median [IQR], 17 (13–23) days vs. 15 (12–20) days; *p* = 0.048) than those in the LR group. No differences in blood loss, blood transfusion rate, blood transfusion volume, Pringle maneuver rate, ICU stay rate, and hospital death rate were found between the two groups.

With respect to postoperative complications, the CLR group had significantly higher incidence rates of intra-abdomen bleeding (17.6% vs. 8.1%; *p* = 0.046), biliary leakage (14.7% vs. 4.8%; *p* = 0.018), PHLF (20.6% vs. 9.7%; *p* = 0.035), overall complications (44.1% vs. 21.0%; *p* = 0.001), and Clavien–Dindo grade IIIa complications (14.7% vs. 3.2%; *p* = 0.003) than the LR group. The incidence rates of severe ascites, abdominal infection, inferior vena cava thrombosis, intestinal obstruction, pulmonary infection, pleural effusion, pulmonary embolism, and hemorrhagic shock were comparable between the CLR and LR groups.

The postoperative pathological features of the two groups were also compared. Patients in the CLR group had a significantly lower risk of developing microvascular invasion (MVI) (26.5% vs. 67.7%; *p* < 0.001) and satellites (10.3% vs. 24.2%; *p* = 0.020) than those in the LR group. The two groups showed no significant difference in lymph node metastasis, invasion of adjacent organs, and cancer-free resection margin. Due to the complete or partial necrosis of tumors in some patients following conversion therapy, histological grading information could not be obtained from the tumor specimens of 16 (23.5%) patients in the CLR group.

### 3.3. Survival Outcomes

The median follow-up period for OS was 15.2 (IQR, 9.6–22.0) months in the CLR group and 23.7 (IQR, 15.2–33.4) months in the LR group. Median OS for both the CLR group and the LR group has not been reached (Figure 1A). The estimated 1-year, 2-year, 3-year, and 4-year OS rates were 88.9%, 82.9%, 82.9%, and 55.3% for the CLR group vs. 87.5%, 72.9%, 65.5%, and 60.5% for the LR group (HR 0.724; 95% CI 0.356–1.474; *p* = 0.374; Figure 1A), respectively. The median follow-up period for RFS was 11.3 (IQR, 7.0–17.0) months in the CLR group and 19.5 (IQR, 7.2–28.1) months in the LR group. Median RFS was 17.2 (95% CI, 11.2–23.2) months in the CLR group and 24.6 (95% CI, 17.1–32.1) months in the LR group (Figure 1B). The estimated 1-year, 2-year, and 3-year RFS rates were 63.5%, 32.7%, and 32.7% in the CLR group vs. 65.3%, 51.7%, and 40.5% in the LR group (HR 1.249; 95% CI 0.807–1.934; *p* = 0.374; Figure 1B), respectively.

### 3.4. Subgroup Analyses

In the subgroup analyses, no differences were found in treatment outcomes based on age, sex, liver cirrhosis, tumor number, tumor size, and AFP (Figure 2). Survival outcomes in HCC patients based on BCLC stage A/B/C were compared. The subgroup analyses showed that the OS of the CLR group vs. the LR group was significantly better among patients with BCLC stage C (HR 0.334; 95% CI 0.116–0.960; *p* for interaction = 0.017; Figure 2A). For patients with BCLC stage A, a significantly worse RFS was observed in the CLR group vs. the LR group (HR 3.643; 95% CI 1.366–9.721; *p* for interaction = 0.005; Figure 2B).

## 4. Discussion

In this study, we compared the perioperative data and long-term survival outcomes between the CLR group and the LR group after PSM. Although the CLR group was associated with longer operation time, longer hospital recovery time, and a higher incidence of postoperative complications, no significant statistical differences in OS and RFS were observed between the two groups. Additionally, a comparison of postoperative pathological features revealed that the CLR group had lower incidence rates of microvascular invasion and satellites than the LR group. Upon subgroup analyses, the CLR group was found to be associated with significantly better OS in BCLC stage C HCC, but significantly worse RFS in BCLC stage A HCC, compared to the LR group.

In China, the majority of HCC patients are diagnosed at an intermediate to advanced stage. It was reported that some HCC patients who initially did not meet the criteria for surgery or liver transplantation, after undergoing conversion or downstaging therapy, could achieve comparable survival outcomes to those initially eligible for liver resection or transplantation [19,20]. However, the majority still failed to meet the criteria for liver transplantation after conversion therapy, due to our selection of patients with more advanced HCC patients for triple combination conversion therapy. While a small percentage of patients were suitable candidates for liver transplantation after conversion therapy in our medical center, the scarcity of liver grafts in China and the high cost of transplantation made liver resection the preferred treatment option.

Conversion therapy for advanced HCC based on various approaches and regimens is being studied, and there is no consensus on which conversion treatment option is the best. In the selection of a conversion treatment option, the ORR and the successful conversion rate are important indicators [15]. The use of single-agent TKIs or ICIs yields limited clinical results, with an ORR ≤ 20%, due to the high heterogeneity and complex pathogenesis of HCC [21,22,23,24]. Several studies have proven that the combination of TKIs and ICIs may yield a synergistic antitumor effect and achieve a higher ORR (approximately 30%) [7,8]. However, the ORR and the proportion of patients who may benefit from conversion therapy have remained far from satisfactory. The triple combination conversion therapy of locoregional treatment with TKIs and ICIs was developed to further improve treatment efficacy, and could lead to an ORR of approximately 60% and a conversion surgery rate of 20~40% [12]. While high ORR and conversion resection rates are positive indicators, the relative safety of conversion treatment strategies should also be considered. With the current data, approximately 30~70% of grade 3–4 adverse events occurred in patients who received the triple combination therapy of TACE/HAIC plus TKIs and ICIs [11,12,25,26,27]. Several studies have proven that the triple combination therapy of TACE/HAIC combined with TKIs and ICIs may not increase the incidence rate and severity of AEs compared with TKIs + ICIs [12,27].

Despite the triple combination conversion therapy strategy showing a promising ORR, a successful conversion rate, and acceptable treatment-related AEs, the safety and long-term survival outcomes of salvage liver resection in patients with initially unresectable HCC who regained the opportunity for surgery remains to be investigated. Although previous studies with small samples have demonstrated the safety of surgery after conversion therapy with TKIs plus ICIs [28,29], the limited availability of quality evidence along with the additional application of locoregional treatment may undoubtedly raise safety concerns regarding the implementation of sequential liver resection following triple combination conversion therapy. Luo et al. found that patients who received surgery following triple combination conversion therapy had more blood loss, longer operative time, higher blood transfusion rates, and longer hospital stays than those patients who underwent hepatectomy alone [13]. Zhang et al. also demonstrated that the conversion surgery group (80.8% patients received conversion therapy of locoregional therapies + TKIs + PD-1 inhibitor) was associated with longer operation time, more blood loss, longer postoperative hospital stay, and longer abdominal drainage time [30]. In the current study, patients who received conversion surgery had a longer Pringle maneuver time, longer operation time, and longer hospital stays than those who underwent pure liver resection, although there were no significant differences in the blood loss, blood transfusion rate, and blood transfusion volume. This result might be related to the preoperative TACE/HAIC, which may result in hepatic tissue inflammation and perihepatic tissue adhesion, which could in turn render liver resection more difficult [31,32]. It was assumed that conversion surgery may be associated with more intraoperative blood loss due to the bone marrow suppression caused by preoperative chemotherapy drugs and the antiangiogenic effect of TKIs [33]. However, previous studies showed that sufficient discontinuation time of TACE/HAIC and TKIs before surgery may not lead to an increase in blood loss [33,34,35]. The current study demonstrated that conversion surgery after a sufficient “withdrawal period” may not increase the risk of intraoperative bleeding, although there may still be an association with more complicated surgical circumstances.

In addition, we found that patients who received conversion surgery had significantly higher incidence rates of intra-abdominal bleeding, biliary leakage, PHLF, overall complications, and Clavien–Dindo grade IIIa complications. We speculate that the increased risk of postoperative intra-abdominal bleeding and biliary leakage may be related to liver function damage caused by conversion therapy and chemotherapy-related hepatic sinusoidal injury [35,36,37]. Furthermore, the complex surgical circumstance and the long operative time may also increase the risk of postoperative intra-abdominal bleeding and biliary leakage [38,39]. In the current study, patients who received surgery following triple combination conversion therapy were associated with a higher rate of PHLF (20.6% vs. 9.7%; *p* = 0.035). The underlying mechanism remains unclear, but the potential hepatotoxicity of conversion therapy may contribute to the occurrence of PHLF in a few ways: (1) postoperative TACE/HAIC may induce chemotherapy-associated liver injuries, including sinusoidal obstruction syndrome (SOS), nodular regenerative hyperplasia (NRH), steatosis, and steatohepatitis, facilitating the occurrence of PHLF [33,36] and (2) TKI-related hepatic toxicity and immune hepatitis caused by ICIs may also be a potential cause [40,41]. While patients who underwent conversion surgery exhibited a higher occurrence rate of PHLF, it is worth noting that the majority of these cases were classified as class A/B (13/14, 92.9%), and no deaths related to PHLF were observed. Additionally, although there was a higher incidence rate of overall complications and Clavien–Dindo grade IIIa complications in patients who underwent conversion surgery, postoperative safety was manageable through appropriate medical interventions.

Additionally, no significant statistical differences were observed in OS and RFS between the conversion resection group and the pure resection group. This study further validates the benefits of surgical resection following conversion therapy. Subgroup analyses showed that conversion surgery was associated with significant improvements in OS for HCC patients with BCLC stage C, but such surgery was associated with significantly worse RFS in BCLC stage A HCC patients. For patients with BCLC stage A disease, the unresectability was mainly attributed to the “surgically unresectable”, such as insufficient postoperative remnant liver volume, and close tumor proximity to the crucial intrahepatic vessels. Although these patients regained the opportunity for liver resection by conversion therapy, they tended to have more risk factors of recurrence than initially resectable HCC patients. While hepatectomy is not recommend for BCLC stage C HCC by the European Association for the Study of Liver Disease guidelines or by the American Association for the Study of Liver Disease guidelines, it was found that a portion of patients with BCLC stage C HCC may obtain a better survival benefit through hepatectomy than through TACE or systemic treatment [42,43,44]. Therefore, liver resection is applied to some cases of BCLC stage C HCC in clinical practice in China, in accordance with the Chinese guideline recommendations [45]. As the current study demonstrated, the conversion therapy, as an important treatment strategy, may provide a more favorable survival benefit for patients with advanced stage HCC. Furthermore, lower incidences of MVI and satellite nodules were identified in the postoperative pathology of the conversion resection group. This may suggest that preoperative neoadjuvant/conversion therapy can potentially inhibit postoperative dissemination and metastasis of tumor cells. However, there were no observed survival benefits in the conversion surgery group, as the initially resectable HCC may represent a more biologically favorable group.

Several limitations that exist in this study should be noted. Firstly, although the PSM method was used to reduce potential bias, selection bias remains unavoidable due to the study’s retrospective design. Secondly, there is no consensus on the choice of conversion therapy regimens, cycles, resectable criteria, and appropriate operation time. The heterogeneity of conversion therapy may diminish the reliability of the study results. Lastly, as the majority of studies focusing on triple combination conversion therapy were performed in Eastern Asian countries, it remains unclear whether the different demographic characteristics between Eastern and Western countries will influence the effectiveness of conversion therapy.

## 5. Conclusions

In conclusion, although hepatectomy following triple combination conversion therapy resulted in longer operation time, longer hospital recovery time, and a higher incidence of postoperative complications, the perioperative safety profile is acceptable with appropriate medical adjustments or interventions. Furthermore, liver resection following triple combination conversion therapy was associated with favorable survival outcomes. Our study demonstrated that liver resection may be a safe and effective treatment option for patients with initially unresectable HCC who regained the opportunity for surgery after triple combination conversion therapy.

## Figures and Tables

**Figure 1 cancers-15-05878-f001:**
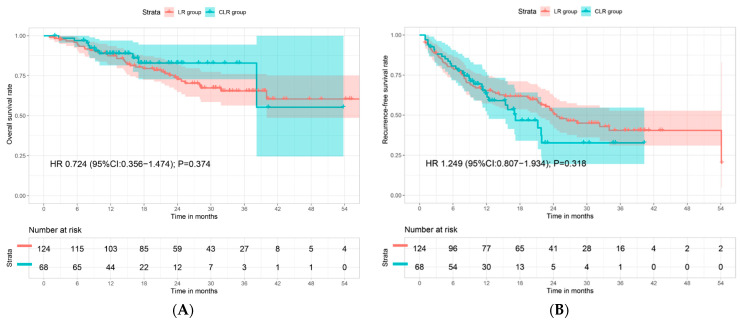
Survival curves of overall survival (**A**) and recurrence-free survival (**B**) after propensity score analysis. CLR = liver resection after conversion therapy; LR = liver resection; HR = hazard ratio.

**Figure 2 cancers-15-05878-f002:**
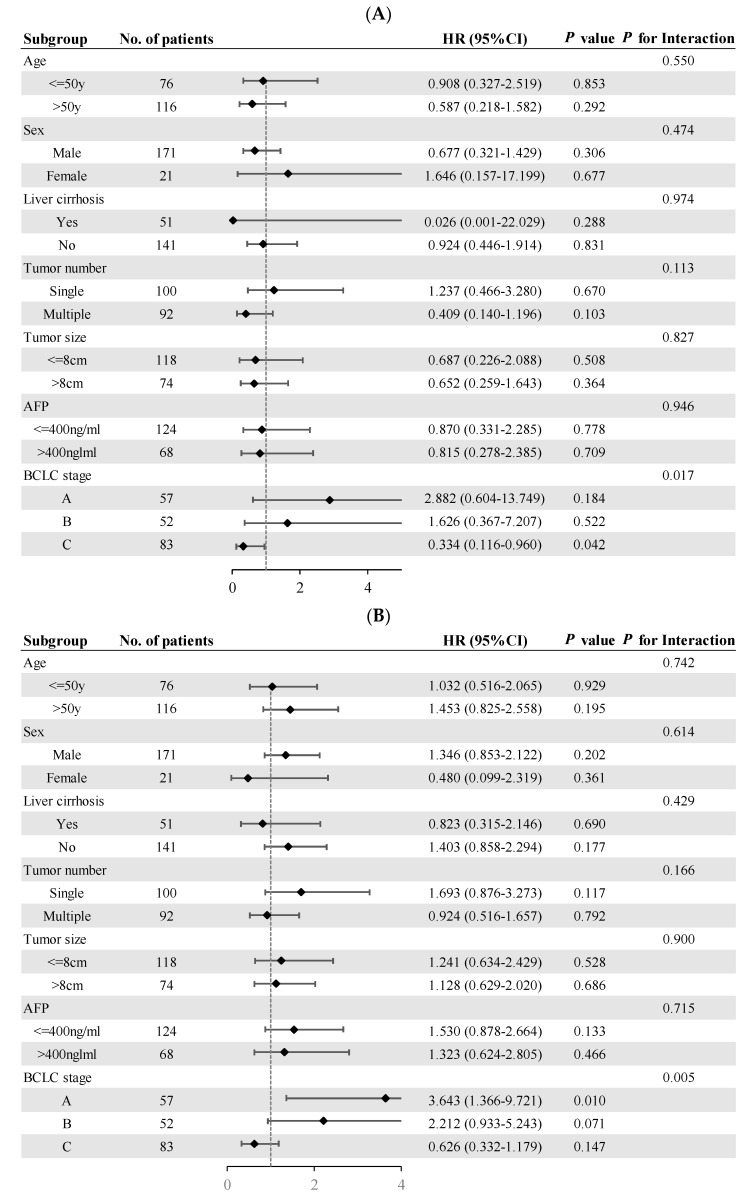
Subgroup analyses for overall survival (**A**) and recurrence-free survival (**B**).

**Table 1 cancers-15-05878-t001:** Preoperative baseline characteristics of study patients before and after propensity score analysis.

	Before Propensity Score Matching				After Propensity Score Matching			
Variable	CLR Group (n = 75)	LR Group (n = 651)	*p* Value	SMD	CLR Group (n = 68)	LR Group (n = 124)	*p* Value	SMD
Age (years)	53 (43–59)	55 (48–64)	0.015	0.309	55 (48–60)	53 (47–61)	0.865	0.026
Sex (M:F)	67:8	552:99	0.380	0.136	61:7	110:14	1.000	0.032
HBsAg positive	70 (93.3%)	536 (82.3)	0.024	0.341	63 (92.6%)	113 (91.1%)	0.927	0.056
HCVAb positive	1 (1.3%)	7 (1.1%)	1.000	0.024	1 (1.5%)	0 (0.0)	0.760	0.173
Liver cirrhosis	20 (26.7%)	187 (28.7%)	0.811	0.046	17 (25.0%)	34 (27.4%)	0.848	0.055
Child-Pugh grade			0.477	0.123			1.000	0.010
A	71 (94.7%)	596 (91.6%)			64 (94.1%)	117 (94.4%)		
B	4 (5.3%)	55 (8.4%)			4 (5.9%)	7 (5.6%)		
Tumor location			0.607	0.157			0.769	0.044
I	0 (0.0)	7 (1.1%)			0 (0.0)	0 (0.0)		
II~VI	20 (26.7%)	187 (28.7%)			20 (29.4%)	34 (27.4%)		
V~VIII	55 (73.3%)	457 (70.2%)			48 (70.6%)	90 (72.6%)		
Tumor number			<0.001	0.678			0.782	0.065
Solitary	34 (45.3%)	499 (76.7%)			34 (50.0%)	66 (53.2%)		
Multiple	41 (54.7%)	152 (23.3%)			34 (50.0%)	58 (46.8%)		
Tumor size (cm)	7.4 (4.3–9.7)	5.0 (3.0–8.5)	0.038	0.285	7.3 (4.2–9.5)	6.8 (3.4–10.2)	0.961	0.008
Operation approach			0.129	0.200			1.000	0.016
Laparoscopic	36 (48.0%)	377 (57.9%)			34 (50.0%)	63 (50.8%)		
Open	39 (52.0%)	274 (42.1%)			34 (50.0%)	61 (49.2%)		
AFP (ng/mL)	10.6 (3.7–243.7)	62.8 (6.3–1449.0)	0.278	0.145	9.2 (3.6–136.3)	147.5 (10.5–2370.5)	0.473	0.111
BCLC			<0.001	0.880			0.982	0.029
0	0 (0.0)	59 (9.1%)			0 (0.0)	0 (0.0)		
A	21 (28.0%)	362 (55.6%)			20 (29.4%)	37 (29.8%)		
B	18 (24.0%)	109 (16.7%)			18 (26.5%)	34 (27.4%)		
C	36 (48.0%)	121 (18.6%)			30 (44.1%)	53 (42.7%)		

Data are shown as median (IQR) or n (%). Abbreviations: IQR, interquartile range; CLR, liver resection after conversion therapy; LR, liver resection; SMD, standardized mean difference; HBsAg, hepatitis B surface antigen; HCVAb, hepatitis C virus antibody; AFP, alpha-fetoprotein; BCLC, Barcelona Clinic Liver Cancer stage.

**Table 2 cancers-15-05878-t002:** Conversion therapy characteristics.

Variable	CLR Group (n = 68)
TACE	16 (23.5%)
TACE cycle	2 (2–4)
HAIC	52 (76.5%)
HAIC cycle	4 (3–4)
TKIs + ICIs regimen	
Lenvatinib + pembrolizumab	11 (16.2%)
Lenvatinib + tislelizumab	9 (13.2%)
Lenvatinib + camrelizumab	19 (28.0%)
Lenvatinib + sintilimab	3 (4.4%)
Donafenib + tislelizumab	12 (17.6%)
Donafenib + camrelizumab	3 (4.4%)
Apatinib + camrelizumab	11 (16.2%)
Duration of TKIs treatment (days)	100 (72–142)
ICIs treatment cycles	4 (3–4)
Conversion time (months)	126 (92–164)
Tumor response according mRECIST criteria	
CR	18 (26.5%)
PR	45 (66.2%)
SD	5 (7.3%)

Data are shown as median (IQR) or n (%). Abbreviations: CLR, liver resection after conversion therapy; IQR, interquartile range; TACE, transcatheter arterial chemoembolization; HAIC, hepatic arterial infusion chemotherapy; TKIs, tyrosine kinase inhibitors; ICIs, immune checkpoint inhibitors; mRECIST, modified Response Evaluation Criteria in Solid Tumors; CR, complete response; PR, partial response; SD, stable disease.

**Table 3 cancers-15-05878-t003:** Intraoperative and postoperative data of study patients after propensity score analysis.

	CLR Group (n = 68)	LR Group (n = 124)	*p* Value
Blood loss (mL)	400 (163–838)	300 (100–600)	0.123
Blood transfusion	22 (32.4%)	31 (25.0%)	0.276
Blood transfusion volume (mL)	0 (0–400)	0 (0–150)	0.225
Pringle maneuver	54 (79.4%)	87 (70.2%)	0.165
Pringle maneuver time (min)	45 (11–76)	30 (0–55)	0.013
Operation time (min)	268 (215–349)	230 (180–300)	0.037
ICU stay (days)	3 (4.4%)	7 (5.6%)	0.713
Hospital death	0	0	1.000
Hospital stays (days)	17 (13–23)	15 (12–20)	0.048
Pathological Features			
Microvascular invasion	18 (26.5%)	84 (67.7%)	<0.001
Satellites	7 (10.3%)	30 (24.2%)	0.020
Lymph node metastasis	1 (1.5%)	2 (1.6%)	0.939
Invasion of adjacent organs	6 (8.8%)	12 (9.7%)	0.846
Cancer-free resection margin	64 (94.1%)	107 (86.3%)	0.097
Histology			<0.001
Well-differentiated	2 (2.9%)	6 (4.8%)	
Moderately differentiated	32 (47.1%)	67 (54.0%)	
Poorly differentiated	18 (26.5%)	51 (41.1%)	
Not available	16 (23.5%)	0	
Postoperative complications			
Intra-abdominal bleeding	12 (17.6%)	10 (8.1%)	0.046
Biliary leakage	10 (14.7%)	6 (4.8%)	0.018
Severe ascites	2 (2.9%)	1 (0.8%)	0.254
Abdominal infection	0	2 (1.6%)	0.540
Inferior vena cava thrombosis	1 (1.5%)	0	0.354
Intestinal obstruction	1 (1.5%)	2 (1.6%)	0.939
Pulmonary infection	1 (1.5%)	2 (1.6%)	0.939
Pleural effusion	6 (8.8%)	11 (8.9%)	0.991
Pulmonary embolism	2 (2.9%)	1 (0.8%)	0.254
Hemorrhagic shock	1 (1.5%)	1 (0.8%)	0.665
PHLF	14 (20.6%)	12 (9.7%)	0.035
Overall complications	30 (44.1%)	26 (21.0%)	0.001
Clavien-Dindo grade			
IIIa	10 (14.7%)	4 (3.2%)	0.003
IIIb	2 (2.9%)	2 (1.6%)	0.538
IVa	2 (2.9%)	5 (4.0%)	0.700
IVb	1 (1.5%)	0	0.354

Data are shown as median (IQR) or n (%). Abbreviations: IQR, interquartile range; CLR, liver resection after conversion therapy; LR, liver resection; ICU, Intensive Care Unit; PHLF, post-hepatectomy liver failure; Severe complications indicate complications of Clavien–Dindo grade IIIa~V.

## Data Availability

The data presented in this study are available on request from the corresponding author. The data are not publicly available to safeguard patient privacy.

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
