# Peer review of "Safety and Survival Outcomes of Liver Resection following Triple Combination Conversion Therapy for Initially Unresectable Hepatocellular Carcinoma"

_cancers, 2023, doi:10.3390/cancers15245878_

Round 1

Reviewer 1 Report

Comments and Suggestions for Authors

Liver resection is depicted as the best therapy for small HCC. Liver transplantation (LT) is not mentioned. Recurrence rates and survial are not in favor of resection for HCC in cirrhosis staged UNOS T2 or succesfully down-staged HCC. see Reig-M 2022. (Would LT has been an option for some of the patients? The rate or cirrhosis appears to be very low with 25%? Was portal hypertension a criteria for decision?). FU is very short. Recurrence-free survival analysis should have a longer FU period.

Author Response

Response:

As the reviewer has stated, HCC patients undergoing downstaging to within United Network for Organ Sharing T2 criteria before liver transplantation were associated with comparable 5-year post-transplant recurrence-free survival and overall survival rates compared to those meeting T2 criteria without downstaging (Francis Y Yao 2015). However, the majority still failed to meet the criteria for liver transplantation after conversion therapy, due to our selection of patients with more advanced HCC patients for triple combination conversion therapy. While a small percentage of patients were suitable candidates for liver transplantation after conversion therapy, the scarcity of liver grafts in China and the high cost of transplantation made surgical resection the preferred treatment option. (Please see the revised manuscript, page 9, second paragraph).

The diagnosis of cirrhosis is based on the presence of nodularity, splenomegaly, or portal hypertension observed in ultrasonography, computed tomography, or magnetic resonance imaging, which may contribute to a lower diagnostic rate. Portal hypertension is a consideration standard in our treatment decision-making process. Portal hypertension is not regarded as an absolute contraindication to liver resection in HCC patients with compensated cirrhosis (Takeaki Ishizawa 2008). With improvement in surgical techniques and careful preoperative evaluation, we found that liver resection is a safe and effective treatment strategy for HCC patients with portal hypertension and preserved liver function. However, for patients with concurrent portal hypertension and decompensated liver function, liver transplantation is a more prioritized treatment option given the worse survival rate following hepatectomy (Roberto Santambrogio 2013). In addition, we do not consider such patients suitable for triple combination conversion therapy due to concerns about the occurrence of severe adverse events. If patients experience the symptoms of decompensated liver function (such as elevated transaminases, elevated bilirubin levels et al.) during the conversion therapy process, we contemplate adjusting the drug dosage or discontinuation until organ function improves. If patients meet the Milan/UCSF criteria and exhibit signs of decompensated liver function or significant symptoms of portal hypertension after conversion therapy, we recommend proceeding with liver transplantation.

As for the reviewer’s second point, we do appreciate the reviewer's insights. However, due to the relatively short duration of the study on the triple combination conversion therapy, it is inevitable that the follow-up period was limited. We plan to continue this study in the future, confident that subsequent research will yield more reliable survival outcomes.

Reviewer 2 Report

Comments and Suggestions for Authors

This is very good paper about the patients who received hepatectomy after conversion therapy.

Abstract is good, the introduction is satisfactory written, results very good  presented and organized, methods part is well explained, discussion is good and conclusion is clearly written.

Author Response

Thank you very much for your recognition of our work. Here, on behalf of all the authors, I would like to express my heartfelt thanks to all the authors and wish you greater success in your work.

Reviewer 3 Report

Comments and Suggestions for Authors

Dear Editor and Authors,

It was my pleasure to review this manuscript titled “Safety And Survival Outcomes of Liver Resection Following Triple Combination Conversion Therapy for Initially Unresectable Hepatocellular Carcinoma” by Dr. Long and colleagues from the Department of Hepatobiliary Surgery, Sun Yat-sen Memorial Hospital of the Sun Yat-sen University in Guangzhou, China.

In this retrospective, single institution analysis the authors present their experience and outcomes in utilizing triple combination conversion therapy with tyrosine kinase inhibitors (TKIs), immune checkpoint inhibitors (ICIs) and hepatic arterial infusion chemotherapy (HAIC) in unresectable hepatocellular carcinoma.

This is a well conducted study with clearly established parameters both in terms of inclusion and exclusion criteria as well as a extensive and appropriate statistical analysis. It is important that the authors have conducted propensity score matching  to investigate and reduce potential co-founding variables. Consequently, the results of the study are robust and interesting.

The manuscript is well written, in concise and easily understood language with only minor editing required. It is also well illustrated with tables and figures.

As a result, I am happy to accept this manuscript for publication as is. Congratulations to the authors as this is not something I easily do!!

Comments on the Quality of English Language

Only minor editing is required which can be done at proofreading.

Author Response

Thank you very much for your recognition of our work. Based on your suggestions, we invited a native English-speaking professor to revise our paper. On behalf of all the authors, I would like to express my sincere gratitude to you and wish you continued success.